# Bodily Practices and Meanings Articulated in the Physical Exercise of Older Adults in Santiago de Chile Post-COVID-19

**DOI:** 10.3390/ijerph21050567

**Published:** 2024-04-29

**Authors:** Alexis Sossa Rojas

**Affiliations:** Institute of Sociology (ISUC), Pontifical Catholic University of Chile, Santiago 7820436, Chile; apsossa@uc.cl

**Keywords:** Santiago de Chile, older adults, physical exercise, COVID-19, qualitative research

## Abstract

This article presents the results of almost nine months of ethnographic research on the relationship between physical exercise and health in older people in the post-COVID-19 context. Via exploratory–descriptive qualitative research and the use of a convenient sample, I shed light on this relationship using the stories and life experiences of 40 older people (10 men and 30 women, including two women instructors for senior classes) who exercise regularly. The meanings they attributed to physical exercise during COVID-19 and after it are explained, emphasising first that there is no health in a context of not feeling safe; once there is a feeling of security, the most relevant meanings can be exposed in three directions. First, exercise produces a sense of identity linked to “being an athlete” and “belonging to a group”. Second, exercising is valued as participating in something meaningful (the meanings range from self-realisation, independence, and autonomy to feelings of happiness). Finally, and linked to the sense of identity, those who train alone show more commitment and total hours spent in physical exercise and physical activity than those who train in groups. Even though older people are not a homogeneous group, they generally faced the pandemic as an ageist situation that affected their lives and how they saw sports and health. This article describes the strategies they used during COVID-19 related to exercise and well-being and those used once the pandemic restrictions were no longer present. The qualitative aspects that physical exercise brings to this population are highlighted. The research results give voice to older people, showing their heterogeneity and the meanings and practices that unite them. These inputs are rich material for studies on physical activity, older people, and well-being.

## 1. Introduction

Older adults are not a homogeneous group [1], and even though ageing is a multidimensional, progressive, intrinsic, and universal phenomenon [2]., not all people age or perceive old age in the same way. Nevertheless, four phenomena regarding ageing are widely discussed in the scientific literature. First, ageing populations will be one of the most significant social transformations of the twenty-first century worldwide [3]. (Second, with advancing age, older people tend to reduce their levels of participation in social and physical activities [4,5,6]. Third, physical inactivity is the fourth leading risk factor for global mortality and is estimated to cause 6% of deaths worldwide. Physical inactivity is increasing; it has increased disease prevalence and poses a substantial economic burden globally [7]. Finally, there is an inclination to ignore older people’s perceptions and beliefs about the way in which ageing is experienced [8,9,10,11] and, in particular, the role and meaning of physical activity in later life [12,13].

The experience of COVID-19 since 2020 has significantly impacted the already complex ageing phenomenon. Impositions such as social distancing have been proven to affect the mental health of older adults, and mental ill-health is an independent risk factor for depression, anxiety disorders, suicide, and the worsening of existing psychiatric symptoms, in addition to increasing feelings of fragility, loneliness, stress, and irritation [14]. Similarly, evidence supports the hypothesis that quarantine harms psychological health and causes issues, including post-traumatic stress symptoms, confusion, and anger [15,16]. For instance, in a study that involved clinically stable older adults during COVID-19, the prevalence of depressive and anxiety symptoms was found to be 62.3%, which suggested that the outbreak of COVID-19 had a significant impact on the mental health of those individuals [17].

In addition to isolation or social distancing that tend to occur as one ages, age itself brings such phenomena as sarcopenia. The European Working Group on Sarcopenia in older people defined sarcopenia as a progressive and generalised musculoskeletal disorder which intensifies after the age of 50, with a 1.5–5% annual loss in muscle strength, associated with a higher probability of risk of falls, fractures, physical disability, and mortality [18]. For all this, and with physical exercise being one of the most critical elements to help older people live well, it is essential to study the relationship between physical exercise and health and investigate how COVID-19 affected or not this relationship.

In this article, I consider how older people who train frequently have been resuming their lives since the restrictions and measures that were taken during the pandemic ended and, retrospectively, how this population lived through the pandemic. This article brings together the experiences of 40 older people (10 men and 30 women, including two women instructors for senior classes) who exercise regularly to better understand the connections between physical exercise and health.

The data were collected using the application of an ethnographic methodology in Santiago, Chile’s capital, from May 2022, when the restrictions and the pandemic were still present, until the end of January 2023, when the virus was causing fewer problems and restrictions. Chile, in general, and Santiago, in particular, are useful in understanding the relationship between physical activity and the ageing phenomenon for four main reasons.

First, Chile, Cuba, and Uruguay are among those countries that have the most rapidly ageing populations, with older adults projected to comprise more than 20% of Chile’s population by 2025 [19]. During the last 40 years, Chile’s older adult population has tripled; this figure is in line with data that indicate that it is the only country in Latin America with an average life expectancy of more than 80 years. Similarly, by 2050, Chile will have the highest proportion of older adults in Latin America. This increase will be 109.5% on 2020 figures, exceeding the 74.7% projected for the world population [3].

Second, Chile was one of the Latin American countries more affected by COVID-19 [20,21,22,23]. It is considered one of the countries with the worst outcomes in terms of the number of cases and death rates [24]. Yet, as soon as the COVID-19 restrictions eased (or even at the height of the pandemic), my informants carried out physical activity.

Third, Chile has very high figures for sedentary lifestyles [25]. The Chile National Health Survey estimates that 86.7% of the population is sedentary and that around 75% of the Chilean population is overweight. This makes my informants a particular population that can shed light on the relationship between exercise and older people in largely sedentary contexts.

Finally, the pandemic began in Chile after the October 2019 rebellion, which was an extensive political mobilisation. During this period of political unrest, protesters took to the streets to demand structural changes in the political and economic systems. This process greatly impacted daily life, public transport access, and street-free movement [26].

For these reasons, Chile offers the opportunity for researchers to deepen their understanding of the relationship between levels of physical activity and ageing (during and after COVID-19). The capital of Chile, Santiago, is home to more than 40% of the population and was the area most severely impacted by the virus. There, the lockdowns were the most severe and lengthy compared with the rest of the country, and the most restrictions on people’s mobility were implemented [27,28]. In addition, Santiago is a city with a larger-than-average older adult population.

## 2. Materials and Methods

The chosen methodology was a qualitative cross-sectional study that used a convenient sample. Qualitative research is sensitive to the complexity of ageing, sport, the body, and society [29,30]. By focusing on the stories that people are told and the stories older people live and tell about themselves and their bodies, we come to understand how people affect their own experiences and make sense of their actions [31]. Personal stories about COVID-19, exercise, and ageing are socially shaped and embodied, and the goal of the methodological design of this study was to capture those meanings.

### 2.1. Data Collection

The data were obtained using ethnography. I sought to establish a sound rapport with collaborators to understand exercise and ageing phenomena better. Despite the shadow of the virus, I spent many hours with my respondents, and this investment of time won me their trust and provided rich data.

My fieldwork began in May 2022, when the pandemic was beginning to wane in Chile. Given Chile’s high rates of inactivity, it was a challenge for me to find older people who exercised. Additionally, winter was approaching, and in Santiago, in winter, levels of physical activity decreased further [25]. COVID-19 and the regulations designed to reduce or prevent contact with others also led to difficulty finding people to work with. For this reason, most of my collaborators worked at centres for older adults in two municipalities (Ñuñoa and San Joaquín). These administrative units provide various types of support, events, and activities for seniors, either for free or for a small fee. The rest of the collaborators trained privately and had no association with any group (but were residents of San Joaquín or Ñuñoa).

The ethnographic data were obtained over almost nine months using participant observation, field notes, and 38 open interviews. Forty people was the point at which saturation was reached. These collaborators signed consent forms that indicated their interest in participating in my research. I spent months talking with them and sometimes accompanied them to their sports activities. I met with them about three times a month for almost nine months. After this, I kept in touch with many via telephone, text messages, and sporadic meetings.

I conducted in-depth interviews at the end of or close to the end of the nine months of ethnographic work. All were audio-recorded (it was impossible to interview with only two people). The interviews lasted between 50 and 100 min.

Perhaps for many, by January 2023, there was no longer much talk about COVID-19. However, for my informants, it was still very much an issue. Many of their sports activities remained under capacity restrictions. Many people continued to use alcohol gel and a mask and to avoid physical contact with strangers or with objects used by others. For example, despite the weight of their dumbbells, Teresa and her partner brought this equipment to their training activities at a municipal gym.

The criteria used to recruit the participants were that they had to be people over 65 years of age who exercised at least twice a week, with 60 or more minutes dedicated to physical exercise per week (and all my collaborators far exceeded that criterion). Two instructors in sports activities for older adults were included because they specialised in older people and were close to being older people themselves. Accordingly, I worked with 38 people between 65 and 83 years (averaging 72 years) and two instructors, each 63 years old. There were 10 men and 30 women, of whom two were instructors for senior classes.

More details about my collaborators can be seen in the following Table 1.

### 2.2. Data Analysis

A thematic analysis was carried out. MAXQDA software (v12.0) was used to code and generate themes. Since the ethnographic work was long, many relevant themes emerged. Some of them, such as the issue of injuries and the precautions that older people should take when performing physical exercise, have been discussed elsewhere [32]. In this article, I will draw on critical physical practices during the pandemic and the meanings that the participants articulated regarding physical exercise post-COVID.

To ensure the credibility and trustworthiness of the findings, 1.5 months were spent on data familiarisation and organisation, during which period interviews were transcribed, and the field notes were organised. After that, four months were spent on the analysis, and one month was used to triangulate the results via discussion with the informants themselves.

### 2.3. Ethical Considerations

The people who decided to help me were all aware that they were participating in this research; they knew what this research required of them. I explained my research to them and told them that they had the right to withdraw from it whenever they wanted to. They all gave me their signed consent to be part of my study. This research was also approved by the Ethical Committee of the Pontifical Catholic University of Chile.

## 3. Results and Discussion

### 3.1. Bodily Practices of Older Adults in Santiago de Chile during COVID-19

My fieldwork took place when the COVID-19 pandemic was waning and life was returning to normal. Therefore, confinement and strict quarantines were discussed and studied retrospectively. It is relevant to mention the pandemic period because although it had passed by the time of this work, it had been a unique time during which the relationship between exercise and health had been affected, questioned, and rethought. In this section, I delve into topics that came up most often in my fieldwork and interviews.

#### 3.1.1. Without Security, There Is No Health

An important topic that the fieldwork revealed was the issue of security. This covered the measures and practices that older people could or could not put in place to protect themselves (or to feel protected) or their sense of security, such as being able to go into the street and not fall or being able to go out and not be assaulted or discriminated against. For my informants, insecurity directly affected their perception of health and the practices they felt able to undertake that could contribute to its improvement. The literature shows that perceptions of risk and fear are strongly linked to worry, which can evoke negative emotions and elevate levels of anxiety and distress that are related to fears about uncertainties and potential adverse health outcomes [33,34,35,36,37].

Therefore, the relationship between physical exercise and health in older people cannot be understood without referral to how the pandemic process was experienced and how it changed or did not change their practices and decisions.

The pandemic impacted Chile at a particularly complex social and political time, only a few months after the 2019–2020 Chilean protest. It was a period of insecurity, violence and economic decline. Later, the pandemic plunged the economy into the worst recession in decades [38]. Sophia said, “When I had the hardest time, when I was most afraid, was during the social outbreak, not with the coronavirus”. Violeta said something similar,

“The social outbreak caused a lot of fear and uncertainty. Since then, I haven’t gone out at night or not even in the afternoon; I’ve spent more than four years where after 5 p.m., I no longer go out on the street. Because of this, my days are shorter, and if I have something to do, I have to sacrifice my physical exercise. I can’t do both on the same day”.

Here, Violeta expresses some feelings of discomfort and explains that she no longer ventures out after 5 in the afternoon; any activity outside her house must be adjusted to a reduced timeframe. This situation often involves replacing her exercise schedule to carry out other activities.

When COVID struck, my informants were already familiar with concepts such as uncertainty and fear due to the social unrest. These emotions come from perceptions derived from specific personal experiences (such as protests, robberies, violent marches, and street shootings). Still, sociologically, they are also rooted in particular types of social structures, ways of life and frameworks of meanings. For example, the daily news programmes shown on television greatly affected this feeling of fear. This situation continued or worsened with the pandemic and strongly influenced my informants’ well-being.

The first case of COVID-19 in Chile was confirmed on 3 March 2020, and on 18 March, a State of Constitutional Exception of Catastrophe was declared in the territory [39]. This gave the government the power to restrict freedoms of movement and association and to establish actions such as mandatory quarantines, restrictions of visits to long-term care centres, and the closure of all-day centres, clubs, and organisations for older adults (health checks, procedures, and surgeries were also restricted or suspended) [40].

No national lockdown was established, unlike in neighbouring Argentina and Peru, but rather localised lockdowns and a night curfew (10 p.m.–5 a.m.) were applied across the whole territory [23,38]. All these changes contributed to a general feeling of uncertainty and insecurity.

According to my fieldwork, the main inconvenience that most of my collaborators expressed was that the pandemic had locked them in their homes. Emily said, “They locked us up just as if we were criminals.” Many of my research participants described the confinement as arbitrary and unfair and stated that it worsened their physical and mental health. The COVID-19 pandemic enabled the use of age as clear grounds for discrimination. Ageism was one of the most notorious co-occurrences of the pandemic [41,42,43], mainly because the global trend was that countries used age as a criterion for lockdown. This method is ageist because by instructing older people not to leave their homes while the rest of society is engaged in a semi-normal routine, one of the underlying messages is that older people cannot make appropriate decisions to protect themselves.

“I’m a healthy man; I’ve played sports all my life. You leave me without sports, and you kill me; you’re not taking away a hobby; you’re restricting me as a person; you’re limiting my freedom to be myself” (Francisco).

This quote is noteworthy because Francisco practised sports that were required to be performed outside the home: cycling; jogging; and trekking. For Francisco, contact with nature is part of participating in sports. Being forced to stay at home affected him personally and as an athlete. On one occasion, he told me,

“Your accomplishments, your effort of years is lost when you’re locked up; if there is a virus, I can go for a run or ride a bike in isolated places, one can look up how to protect oneself, but they cannot constantly force and scare you to keep you locked up”.

Francisco’s quote explains three critical things. First, it describes how confinement affects sports practice. Second, it affirms that older people are autonomous and responsible adults. Third, the theme of fear is mentioned regarding how bad it was to scare older people continually.

The relevant literature indicates that remaining inactive for one to two weeks during isolation weakens major health determinants such as muscle strength, cardiorespiratory fitness, and use of oxygen during exercise (VO2 max) [44]. Long rest periods in healthy subjects lead to losses of nitrogen, phosphorus, and calcium from skeletal muscle due to inactivity [45]. Symptoms of lack of training over the long term (more than four weeks) include marked decreases in endurance performance, lower lactate thresholds, loss of muscle mass, gradual reduction in muscle force production, and increased injury risk [46,47,48,49]. In other words, an accepted convention is that each week of inactivity causes up to a 10% overall loss in fitness. Other adverse effects of lockdowns include increased body mass and fat percentage, loss of mental sharpness and toughness, insomnia, and depression [50,51].

Francisco was my “most athletic” collaborator, or at least he had competed in the most competitions. His fear of losing his ability as an athlete was acute, but it was not the most prevalent fear among my informants. Undoubtedly, the most significant fear was losing one’s life to the virus.

“Everything that appeared in the news about the virus and older adults was terrifying. I think I suffered from that fear that psychiatrists say, which I think they call cave syndrome, that older adults, or not just them, but when people lock themselves up in a safe space and stay there, they don’t want to leave that space. That happened to me… I went on staying, staying in the house, and then looking for excuses not to go out. I bought everything: bicycles, dumbbells to continue training from home, but I didn’t leave the house for anything… I didn’t want to expose myself to the virus. I had tremendous psychosis” (Gilda).

An interesting particularity in Gilda’s words is that she continued to exercise during the pandemic, which not all my collaborators did, only those most dedicated to sport. Studies have shown that the presence of COVID-19 did not stop athletes, who immediately adapted to the new situation [52] What is more striking is that this quote summarises what my collaborators expressed to me the most: the fear of contracting the virus and losing their lives. Many had significant problems that were linked to being alone. Both my field notes and interviews contained references to many saying that the pandemic made them feel older, vulnerable, and sad. Nevertheless, a distinction that many explained to me, and unlike what Gilda expressed, was that confinement and contracting the virus were different things. They feared becoming sick but did not feel that confinement was good for them.

Studies show that in many cases, confinement and fear translate into depression or great loneliness, which is significantly associated with increased anxiety, stress, memory, sleeping, stomach or bowel problems, feeling down, irritability, a reduced sense of life purpose, and suicidal ideation [1,15,27,40,53,54,55]. In addition, and based on evidence from different countries, the opportunity to salvage social connections via virtual platforms was not very successful, generally speaking, for older adults [56]. However, the effectiveness of virtual platforms has been shown in different online training protocols related to physical performance in older adults, emphasising the powerlessness of utilising this strategy to improve these subjects’ quality of life [57,58]

“With the pandemic, I got terrible depression, pain everywhere, and a disease that no one could identify, but it was nothing more than tension, stress. It’s something that the body cannot handle anymore, and it just has to burst. I felt alone and terrible; that’s how diseases such as cancer are produced” (Laura).

Since Durkheim’s classic study of suicide [59], we have known that isolation and lack of connectedness to others tend to be predictors of morbidity and mortality. Those who broaden their social networks are more socially connected and have greater well-being than those who do not. The perception of physical, psychological, emotional, and social distancing from others can cause a range of negative emotional experiences. Social psychology explains this as a deterioration of the social belonging at the core of what characterises human beings [60].

Although social distancing was an essential measure to mitigate the pandemic, studies have shown that continued isolation and loneliness can be severely detrimental for people [61] and may have multiple deleterious physical and mental consequences [62]. Another significant element is that, although Laura said she had depression, her symptoms or discomforts were not so much manifested in mood or sleep problems as in muscle aches and pains.

It is necessary to mention that there were exceptions among my collaborators. For example, Marcela said,

“The pandemic didn’t affect me much. No, because when we could get out (she is referring to going out with her husband), we would go to the supermarket, order what we wanted, and then pick up everything from the car. We also went for walks, even when it wasn’t allowed, and no one said anything to us. We weren’t afraid. We kept a distance from other people and always wore a mask. Besides, we live close to many green areas, and our house is big, and we moved a lot inside it. We would go out to the garden; the garden is large, and we would sit on the balcony. We always found something to do”.

Marcela’s case is interesting, first because the measure that recommended limiting contact with older adults and the use of social distancing often meant isolation in Chile, since in that country, more than 30% of older adults live on their own or with another older person [23]. For Marcela, the isolation was not total because she had her partner. Second, considering that she and her partner had a good economic situation, a big house, a car, a nearby supermarket, and green areas, her circumstances show how sociodemographic aspects helped people to live through the pandemic (or made living through it impossible). It also shows that many people did not respect the quarantines and decided to leave the house when they wanted, demonstrating heterogeneity in quarantine compliance regarding mobility [63]. Something else in this quote is significant, especially if we compare it with what Francisco told us earlier: it was easy to go for a walk (or to take the car and use phone applications that could show where there were police in order to avoid them), but for Francisco, who wanted to go out and play sports, his choice of activities implied a greater exposure to being seen by the police and receiving a fine.

Within this framework of annoyance at the mandatory confinement and the fear of becoming seriously ill due to COVID-19 (and the fear of losing athletic achievements), a second topic to which my collaborators often referred in our conversations was changes in their diets and the consequences thereof.

#### 3.1.2. Food Is Part of Health

“I gained much weight due to the pandemic. I would get up, have breakfast, sit down, knit, make lunch, sit, knit, and I couldn’t go for a walk, even if I wanted to; at first, I tried to walk to the roundabout of the building, but little by little I stopped moving. So, I ended up staying at home for two years, eating, and not doing much, not doing anything, being sedentary, which affected me a lot” (Carol).

Although some of my collaborators, such as Francisco and Gilda, used exercise to manage the stress of the pandemic, many people managed their stress (or loneliness) with food [64]. Overeating and binge eating can lead to regret, physical discomfort, and weight gain [65]. In Carol’s case, the most damaging change was in her body image; this issue came up gradually during our discussions, perhaps out of fear or embarrassment to confess this to a man half her age and, in some way, a “stranger”. After some months, Carol said, “One understands that at this age, one cannot see oneself as when one was 20, but all the weight I gained transformed me; I look at my body and say, ‘this isn’t me’”.

For Carol, the pandemic resulted in her gaining weight, so her body was foreign to her. That alienated look at her own body was also driven by a feeling of shame and discomfort. Studies have found that for older adults, the lockdown had consequences for their image and intensified the occurrence of ageism [66,67,68,69].

“At first, I didn’t notice the weight; I spent all day in my pyjamas; why would I change my clothes if I didn’t go anywhere? What made me feel terrible was when it started to be difficult for me to bend down properly; my back hurt, and I got tired a lot” (Ariel).

As I have mentioned, the sedentary lifestyle brought about by the pandemic reduced muscular and aerobic capacity; this reduction, added to the weight gain, was sometimes not perceived (for example, when wearing pyjamas or baggy clothes) or not necessarily troublesome because as Leonardo told me, “We all thought that the pandemic was only going to last a couple of months at most”. Ariel noticed it when the change was significant, and it was evident when she had to bend over.

Francisca said,

“There was a time when I felt increasingly tired and exhausted; I started to get very sleepy, I got up to eat, and then came back to bed, and so I either watched television or slept. One day, I told myself, ‘This isn’t normal; I must be sick; something is wrong with me’”.

Ariel and Francisca’s quotes do not speak to us about their images of their bodies but rather about the body’s capacities and the displeasure they felt on seeing how they were “losing themselves”. Let us also think about the relevance of these quotes expressed by people who were in the habit of exercising frequently.

In his study, Taylor mentioned that in Chile, “at the beginning of the vaccine campaign, there was a message from the government that ‘vaccines are on their way so the pandemic will end soon’” ([21] p. 1). Messages like this added to the general uncertainty that made many people, due to stress or other reasons, begin to eat more and develop sedentary lifestyles. Misleading communication may have partly explained people’s behaviour. It was probably not emphasised enough that vaccination was only one weapon in the fight against the pandemic. The vaccines were proven primarily to reduce disease severity and the death rate but not transmission rates [38]. As a result, many respondents relaxed too early and did not follow healthy lifestyles.

“Since I was despondent, listless, and couldn’t sleep at night, the doctor gave me sleeping pills, but the pills started to make me hungry, so I started to overeat” (Úrsula).

Úrsula’s quote shows how inactivity and the pandemic increased the complexity of living locked up. Úrsula, feeling isolated, began to be sad and have sleep problems, so she resorted to sleeping pills, which not only helped her to stay in bed more but also to consume more food (and, as I also found later, alcohol) than necessary. Let me complement this point with a quote from Henry.

“The pandemic worsened my health a lot; it made me act irresponsibly because I consumed more alcohol than I needed. Every day at home, a glass of wine at lunch was a habit I didn’t have when I worked; at most at the weekend, a spontaneous aperitif, but no more than that. With the pandemic, having a glass of wine or something else became routine, and this practice damaged my body seriously”.

Most of my collaborators found that they ate more during the pandemic than they had before, and some, like Úrsula, Ramona, or Henry, told me that they had also increased their alcohol consumption. This was not a general trend among my informants, but some people followed this pattern, and this led to physical and psychological declines. Some literature reviews found that this was common during the pandemic [70]. In my fieldwork, usually, in the conversations with those people who had drunk “more than necessary” during the pandemic, they showed sorrow about this topic, claiming that the increase was very little until it got out of hand, or they used phrases such as that the anxiety of the confinement won them over.

Different studies have indicated that feeling forced to stay indoors (lockdowns, quarantines) is considered a psychological risk factor for consuming more food of poorer quality compared with standard living conditions. This induced changes in nutrition habits and challenged the participants’ balance of food intake and energy expenditure, which resulted in weight gain [71] or increased alcohol consumption. People began to overcome this problem only once pandemic restrictions were eased.

### 3.2. Meanings Articulated Regarding Physical Exercise Post-COVID-19

In the sections above, I have indicated some of the most relevant practices of my collaborators during the COVID-19 pandemic, their levels of physical exercise, and their lifestyles as older people. It is essential to make clear that the pandemic period strongly marked the relationships of my collaborators not only with physical exercise and health issues but with other parts of their lives; some of their friends did not survive the pandemic, and others have only recently begun, in February 2023, to return to older people’s social centres. Therefore, this period made my collaborators modify their practices and habits as they sought to return to a new normality. Thus, this section explores the meanings my collaborators attributed to their exercise practices from a post-pandemic perspective. Although all my collaborators exercised frequently (and many of them participated in the same activities or groups), not all exercised for the same reasons or attributed the same meaning to it.

When I began my search for informants, the first relevant fact was that older people did not participate much in sports, and finding them engaged in sports activities suitable for all ages was somewhat tricky. Many older people prefer activities that are arranged exclusively for them, are at specific hours, or are sports activities that have the support of a municipality (mainly because they are considered more welcoming and affordable). Scientific studies have highlighted a gender gap in participation in sports activities, showing that women are less regularly involved in such activities than men) [72]. This is not in line with my findings, probably because my search for participants was time-limited. Therefore, I prioritised places where I could find groups of older people, and it is easier to find more women than men in these groups, a trend that has also occurred in other studies in Latin America [73].

Nevertheless, my informants who were most committed to exercising and those who spent the most hours training were those who exercised individually, where the activity itself was their reward. In this sense, it is important to highlight that a measure of minutes per week dedicated to exercise can be misleading about how “sporty” a person is, how those minutes are lived, and their meaning. Here are some cases.

“I train from Monday to Friday, and when I say I train, it’s because I train. I don’t go to class to talk like many do”. These words of Pamela’s were said with emphasis and a display of superiority. She clearly distinguished between herself and those who did not take training seriously [74] explained that this perspective could be described by some as having “serious leisure”.

“People who come with a tradition of doing sports are the ones who look for ways to continue doing sports. We know that muscle that’s not used atrophies”. In this quote and the fieldwork, Francisca showed herself to be a reasonably sporty person. She explained that it was vital for her to return to her routine of training, moving, and exercising. Her identity was as an athlete, and her knowledge told her that her body would deteriorate if she did not participate in sports.

Francisco, Pamela, and Gilda shared these ideas. The concept of athletic identity, which has been defined as “the degree to which an individual identifies with the athlete role” ([75] p. 237), is a motivational factor that affects older adults’ dedication to sports. An essential part of this motivation that people express is that their sports activities set them apart from other people (this may be understood as them being “better than the others”) [25,32,76,77,78].

Ariel said, “Being self-sufficient is being able to move on one’s own; that’s why I want to move, exercise, because I don’t want to depend on anyone”. Ariel did not mention sport as something necessarily crucial in itself, but it was essential to keep the body moving and to avoid or delay dependence on someone. Depending on others is a fear that many of my informants share and that they seek to prevent (or delay) at all costs (physical exercise being one of the best-valued strategies).

It has been shown that when successful, exercise to improve muscle strength increases confidence to negotiate steps and other barriers. Older adults who exercise report increased freedom [79] for instance, they can leave the house and travel independently on public transport [80].

César mentioned something similar, “I’m not afraid of death, but I’m terrified of living a bad life, a limited and restricted life. At this age, I know exercise helps to keep me healthy.” In a complementary line, Ramona said, “When you dance, you get energised because you change your mood, you get happy, then you change your mood and that balances you out because it’s awkward to be angry all day. I dance because it makes me happy”.

These quotes indicate that many of my respondents saw regular engagement in deliberate physical activity as engaging in something meaningful that provided a sense of satisfaction and a feeling of being in charge of oneself [81,82]. This was exhibited differently for each individual. For some, insufficient strength prevented them from leading independent lives, and this affected their perceptions of self and the body [82]. For others, the activity itself brought joy and was vital for their well-being. Quite frequently, I heard from my collaborators that sports improved their mood, which is known because of the physiological and biological characteristics that exercising entails [83].

From a different angle, the case of Úrsula is interesting. When her husband died, she was left alone in her house, and gradually, she stopped going out. Then, the protests in Chile took place, marked by much violence, so she did not go out then either, but she began to feel afraid, sad, and sick, and it was evident to her that the isolation was damaging her. One day, she told herself that she had to leave her house; she went to the nearest centre for older adults,

“I arrived, and I told them I wanted to enrol in a workshop. The lady asked me which one, to which I replied, any. She told me ‘we have a quota for sewing, physical training, and digital updating’ … that’s how I started with physical training classes … Then, when some normality returned (referring to after the pandemic), I went to class again, not so much to train as to talk and share again with my friends there”.

This quote does not discuss avoidance of physical pain, being in charge of one’s body, changing body weight, or improving or maintaining personal records. What stands out here is contact with others, the social aspect of sport. Úrsula confided in me that she had told only two friends, and only after a while, the reasons why she had decided to participate in that activity. Úrsula said, “It’s something very personal; you cannot go around telling everyone that I come to classes because here I think less about my husband, because here I don’t miss him”.

It was crucial to go deeper into this. Many of my collaborators participated in a physical training class. This activity is designed for older adults and provided by the municipality or centres for seniors that have the municipality’s support. Various body movements are practised here; some use implements such as canes, balls, and elasticated bands. There are also moments for dancing to energetic music. Curiously, during my visits to this activity, I found four to eight people who went punctually, signed the list, but did not participate in the class; instead, they sat down to see how the other people trained. To my questions about why they did not join in the activity or why they were there frequently, their answers were focused on two reasons. First, those people who put their names on the list had a greater chance of participating in other activities organised by the municipality, particularly the free trips (also highly valued by my informants). Second, many went to talk among themselves and with the class participants in a safe and entertaining space for recreation and sociability. Having that space and those conversations were a factor in their well-being. On one occasion, referring to being an older person, one of those people told me, “Being more and more alone is not pleasant at all”.

This example and Úrsula’s quote make us reconsider the idea of pain and discomfort (since it can be psychological) and how the concept of identity can help us cope in difficult moments. In these cases, identity is not reflected in being an athlete but in belonging to a group, for example, at a centre for seniors (and in particular benefits, such as travelling). The feeling of belonging to something is essential, considering that individuals with high levels of group identity are more likely to experience well-being [84]. Group interactions are vital to modify lifestyle and maintain the changes [85,86]. The literature suggests that people may gain increased access to emotional support by engaging in social activities, such as sports or religious services. Participation in social activities enables people to interact with like-minded individuals who share similar interests in a particular domain [87]. Similarly, a substantial body of research has suggested that social capital is a resource for mental well-being in older adults [88].

“I have a little age difference with my boss, but if you look at him, he’s an older adult. One time, he dropped his pencil cap in the office and bent down to pick it up, which was pathetic. If one day it’s difficult for you to stand up from a chair and you’re 65 years old, it’s tragic. I like exercising; it makes me light; I move quickly; I don’t get tired.” (Francisco).

This quote helps to explain what Francisco and other informants meant when they said that their bodies asked them to exercise or that exercise made them feel more energetic. The biological explanation is that regular exercise spares the heart by reducing the work it must do. This comes about in two ways: the heart beats less frequently with improvement in aerobic fitness, and the energy requirement of the heart muscle is reduced. The systemic arterial blood pressure also falls, and this reduces the heart muscle’s workload. As an extension of this enhancement by exercise, maximal myocardial performance is increased, and this allows the person to undertake more exercise than they could before. Consequently, after training, people of any age can work harder, longer, and with less effort than previously [1,80]

Sport and regular physical exercise reduce mortality risk by about 35%, and older adults with healthy behavioural lifestyles show four times less disability than those who do not exercise and who are obese. Important lifestyle behaviours involve physical exertion, mental activity, and social interaction due to the role of such activities in sustaining and improving health and well-being [89,90,91] Similarly, meta-analyses and systematic reviews have demonstrated the positive impact of exercise on cognitive function, including improvements in balance, physical characteristics, and quality of life [92].

Participation in sports activities is known to be an effective way for older adults to improve the quality of their lives. Here, I have shown how older adults’ involvement in sports relates to many intertwined motivations, such as health status, a previous history of participation in sports, interest in maintaining and developing social connections, and building stronger bodies in order to enjoy a more independent life. The most relevant meanings can be exposed in three directions. First, exercise produces a sense of identity that is linked to “being an athlete” and “belonging to a group”. Second, exercising is valued as engaging in something meaningful (the meanings range from self-realisation, independence, and autonomy to feelings of happiness). Finally, and linked to the sense of identity, those who train alone show more commitment and total hours spent in physical exercise and physical activity than those who train in groups.

## 4. Conclusions

The COVID-19 pandemic created a scenario that has marked how my informants understand their bodies, physical exercise, and health. The pandemic showed that there was little awareness of the relationship between movement, isolation, and health for older adults. It also highlighted the topic of ageism since both the discourse and policy measures during the pandemic were quite ageist, even though much of the discussion concerning older people’s susceptibility was fuelled by good intentions as the aim was to protect older people [93]. An example was the prioritisation of vaccinations for them [94] Still, what happened during the pandemic can be seen as an opportunity to reframe older age as a period of possibilities, to stress the importance of exercising, to think of older people as capable people, and to see the economic and cultural inequalities that beset our societies.

It is vital to protect older adults’ health and respect and support them in complex situations. COVID-19 was not the first and is likely not to be the last pandemic to rage around the globe and disturb human life and activity worldwide [95]. In this regard, it is worth mentioning that older people formed one of the segments most affected by COVID-19; nevertheless, they showed great flexibility during this challenging period, and even though in Chile, some physical and mental health indicators worsened during the pandemic, older adults mobilised resources that could enable them to maintain their quality of life, such as with improved resilience [40].

This article sheds light on the relationship between exercise and older people in Santiago, which is largely a sedentary context. It shows practices and meanings associated with physical exercise by older people who train frequently. One of the findings of my ethnographic work is that one cannot think about preventive health or greater participation of older adults in sports without thinking about the territory in which they can participate in those sports; the access and proximity that older people have, for example, to centres for older adults with spaces for socialisation and training activities.

Similarly, the inactivity and trouble that some older people may experience and that keep them from performing physical exercise, or the joy and satisfaction that others feel while participating in some sports, depend not only on their bodies, their capabilities, and what they say but also on how they feel (the feeling of security being one of the most important). The motives and feelings that provoke their actions are not always easily identified, so the people who work with them or studies that seek to understand the relationship between sport and older people’s well-being should spend extended periods with them and gain their trust. As shown in this study, feelings such as loneliness or grief are not always easy to verbalise.

Older adults face specific challenges in taking part in sports due to poor physical health, lack of appropriate opportunities, and limited access to sports facilities and resources [96]. However, in this research, I have shown that older adults have diverse motives to participate in sports, such as (a) feeling and being part of a community, (b) maintaining or improving health, (3) feeling like being engaged in something meaningful, (4) achieving a more independent life, (5) and even taking advantage of travel opportunities. In this sense, those who work with and study older people as they undertake sports and physical activities must know the continuum of personal meanings that are associated with participation [31] Isolation and confinement of older adults should be reduced as they may cause physical and psychological problems.

Finally, based on the data collected, it is central to highlight that attention paid to the time that older people dedicate to exercise can be misleading regarding how “sporty” they are or how much exercise they are undertaking, as some people spend a 45 min sports activity mainly talking, while others may invest more physical effort. Still, the effort and how much a person does or can do are very personal characteristics that depend on a series of traits such as their skills, stamina, the elasticity of their body, their muscle pain, and the diseases that affect them, among many others.

## Figures and Tables

**Table 1 ijerph-21-00567-t001:** Collaborators.

Pseudonym	Age	Nationality	Sports Activity
Pamela (she/her)	66	Chilean	Physical training, Gym centre. Sponsored by the municipality (SBM)
Jennifer (she/her)	75	Chilean	Physical training (SBM)
Michel (she/her)	71	Chilean	Physical training (SBM)
Ronald (he/his)	67	Chilean	Gym centre, Swimming (SBM)
Henry (he/his)	68	Chilean	Gym centre. Private and unsponsored activity (PUA)
Manna (she/her)	80	Chilean	Physical training (SBM)
María (she/her)	75	Chilean	Physical training (SBM)
Gilda (she/her)	68	Chilean	Gym centre, Cycling (PUA)
Nori (she/her)	74	Chilean	Physical training (SBM)
Leonardo (he/his)	68	Chilean	Physical training (SBM)
Ramona (she/her)	65	Chilean	Physical training (SBM)
Mavel (she/her)	83	Chilean	Pilates, Chi Kung (SBM)
Carlos (he/his)	71	Chilean	Running, Triathlon (PUA)
Graciela (she/her)	65	Chilean	Physical training (SBM)
Vanesa (she/her)	70	Chilean	Physical training (SBM)
Teresa (she/her)	65	Venezuelan	Gym centre, Pilates, Physical training (SBM)
Dominga she/her)	67	Chilean	Pilates (SBM)
Ana (she/her)	80	Chilean	Pilates (SBM)
Ariel (she/her)	67	Chilean	Pilates (SBM)
Francisco (he/his)	67	Chilean	Trekking, Cycling, Running (PUA)
Laura (she/her)	73	Chilean	Pilates (SBM)
Javiera (she/her)	83	Chilean	Pilates (SBM)
Gonzalo (he/his)	68	Chilean	Physical training (SBM)
Marcela (she/her)	70	Chilean	Chi kung, Pilates, Yoga (SBM)
Sandra (she/her)	63	Chilean	Physical training instructor
Carla (she/her)	82	Chilean	Pilates (SBM)
Claudia (she/her)	63	Chilean	Pilates and Chi kung instructor
Francisca she/her)	74	Chilean	Pilates (SBM)
Josefa (she/her)	67	Chilean	Physical training, Golf (SBM)
Federico (he/his)	70	Chilean	Chi kung, Physical training (SBM)
Úrsula (she/her)	73	Chilean	Chi kung, Pilates (SBM)
Carol (she/her)	71	Venezuelan	Gym centre, Pilates, Physical training (SBM)
César (he/his)	70	Chilean	Gym centre (SBM)
Adan (he/his)	82	Chilean	Pilates, Physical training (SBM)
Agustín (he/his)	83	Dutch	Physical training, Chi kung (SBM)
Jasna (she/her)	80	Chilean	Pilates (SBM)
Violet (she/her)	68	Chilean	Chi kung, Pilates (SBM)
Victoria (she/her)	65	Chilean	Pilates (SBM)
Emily (she/her)	75	Chilean	Pilates (SBM)
Sophia (she/her)	78	Chilean	Physical training (SBM)

Own elaboration. (SBM) Sponsored by the municipality; (PUA) Private and unsponsored activity.

## Data Availability

Some data from this research may be shared as required. Other data, such as interviews, are unavailable due to privacy or ethical restrictions.

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
