# Peer review of "Bodily Practices and Meanings Articulated in the Physical Exercise of Older Adults in Santiago de Chile Post-COVID-19"

_ijerph, 2024, doi:10.3390/ijerph21050567_

Round 1

Reviewer 1 Report

Comments and Suggestions for Authors

Dear author,

this study can give an interesting insight into the influence COVID-19 has had on the habits of the elderly population in a specific country, especially on physical activity. However,

some clarification and editing could make the article more complete.

Comment 1:

from the title we would expect the article to deal with the strategies that even the elderly have adopted to keep active after and during the pandemic. However, this is a retrospective study. so it might be better to rephrase a title more appropriate to the contents of the paper.

Comment 2:

either, I don't know if you have this information, but to make the article innovative, it would be appropriate to include specific information about the strategies, if any, adopted by the elderly to continue to be active at home, for example, the role of technology in this regard and the differences in opportunities for use compared with young people.

For example, Amato et al. in two studies demonstrate the effectiveness of administering different online training protocols to maintain social distance, in elderly subjects with Parkinson's disease (DOI:10.3390/ijerph192013022) and subjects with multiple sclerosis (DOI:10.4081/ejtm.2021.9877). Emphasizing the powerlessness of using this strategy during the pandemic to improve these subjects' quality of life and physical performance.

Comment 3

the language used in the study should be impersonal, avoiding first persons and information not strictly necessary in the methods. This would make the work formally more generalizable.

Comment 4:

some references, expressing very general concepts, could be updated as they are very dated e.g. Fentem, 1994; Grant, 2001… (but also others in the text that are about 20 years old)

comment 5:

please specify better in the "conclusion" section what are the practical implications of this study perhaps emphasizing the highlights.

Author Response

For reviewer 1

Dear author,

This study can give an interesting insight into the influence COVID-19 has had on the habits of the elderly population in a specific country, especially on physical activity. However, some clarification and editing could make the article more complete.

First of all, I want to thank you for the time you dedicated to reviewing the article and for your comments on how to improve it.

I'm writing to inform you that I have edited and improved the article. All changes have been highlighted in yellow in the document so they are easily recognisable.

The main changes have been to improve and update the bibliographic references, the abstract, the introduction, and the conclusion, as well as specific changes to better understand some arguments in the result section.

Now, regarding your comments.

Comment 1:

From the title we would expect the article to deal with the strategies that even the elderly have adopted to keep active after and during the pandemic. However, this is a retrospective study. so, it might be better to rephrase a title more appropriate to the contents of the paper.

The title is strategically important, so this suggestion has made me think a lot about changing it to achieve a better result. Now, I don't know how much I should change it.

To integrate the retrospective part, I could say during COVID-19 and post-COVID-19. However, I consider that by only pointing out post-COVID-19, the previous period is already implied. Secondly, part of the article is retrospective, and it is when the informants talk about how they experienced the COVID-19 process. However, based on that, the article also talks about their current sports practices, so I consider that it is not just a retrospective work.

In short, I have not changed the title, but I have no problem writing during COVID-19 and post-COVID-19 (or another suggestion that the reviewer may consider more accurate).

Comment 2:

Either, I don't know if you have this information, but to make the article innovative, it would be appropriate to include specific information about the strategies, if any, adopted by the elderly to continue to be active at home, for example, the role of technology in this regard and the differences in opportunities for use compared with young people.

For example, Amato et al. in two studies demonstrate the effectiveness of administering different online training protocols to maintain social distance, in elderly subjects with Parkinson's disease (DOI:10.3390/ijerph192013022) and subjects with multiple sclerosis (DOI:10.4081/ejtm.2021.9877). Emphasizing the powerlessness of using this strategy during the pandemic to improve these subjects' quality of life and physical performance.

I read the first suggested text, but there is no reference to the pandemic or Covid-19. The second text does refer to COVID-19, but the study itself moves away from the focus presented in my article. However, the suggestion of incorporating research that talks about the role of technology in this process has been incorporated through the following text:

Webb LM, Chen CY. The COVID-19 pandemic's impact on older adults' mental health: Contributing factors, coping strategies, and opportunities for improvement. Int J Geriatr Psychiatry. 2022 Jan;37(1):10.1002/gps.5647. doi: 10.1002/gps.5647. Epub 2021 Nov 15. PMID: 34729802; PMCID: PMC8646312.

Comment 3

The language used in the study should be impersonal, avoiding first persons and information not strictly necessary in the methods. This would make the work formally more generalizable.

The research is based on ethnographic work; this research method displays its results from the first person. Likewise, it is customary for the researcher to point out relevant aspects of themself that help to understand how the research was carried out and to make clear how personal characteristics such as age, gender, social class, and nationality may or may not have affected the findings. Consequently, I have kept the writing in the first person to be faithful to the ethnographic tradition.

Comment 4:

some references, expressing very general concepts, could be updated as they are very dated e.g. Fentem, 1994; Grant, 2001… (but also others in the text that are about 20 years old).

It is an excellent suggestion, and the references have been updated. These can be seen both in the bibliography and the texts highlighted in yellow.

Comment 5:

Please specify better in the "conclusion" section what are the practical implications of this study perhaps emphasizing the highlights.

I have done it, which can be seen in the new version of the article.

Once again, your feedback has been instrumental in shaping my work, and I'm grateful for the opportunity to address your concerns. I have taken your suggestions and implemented several changes to strengthen my article. Thank you.

Reviewer 2 Report

Comments and Suggestions for Authors

Dear Researcher,

Its a nice research idea with practical applicatopn. some revisions are recommended as follows: 

1- Abstract

-The abstract begins by stating the duration of the research (nine months) but could benefit from a clearer articulation of the research objectives. It mentions using exploratory-descriptive qualitative research but does not specify the research questions or hypotheses guiding the study.

-The abstract indicates that the study explores the meanings attributed to physical exercise during and after COVID-19, emphasizing safety, belonging, and meaningfulness. However, it could be strengthened by providing specific examples or key findings that illustrate these themes.

2- Introduction

-this section needs to be improved. Incorporating theoretical frameworks or concepts that highlight the necessity of conducting such a study would strengthen the introduction. This could involve outlining theories of aging, physical activity, or health behavior change that guide the research and how these theories will be applied or tested in the study. the following updated references are recommended:

 Faraziani, F., & Eken , Özgür . (2024). Enhancing Cognitive Abilities and Delaying Cognitive Decline in the Elderly through Exercise-based Health Management Systems. International Journal of Sport Studies for Health, 7(2), 13-22. 

Ahmadpour, M., & Rezaei, . M. . (2023). The Effect of Whole Body Electromyostimulation Exercises on Improving Static Balance and Self-Efficacy in the Elderly. Health Nexus, 1(4), 39-47. 

  1. -The introduction thoroughly focus on the impact of COVID-19 on older adults but could more closely tie these impacts to the study’s focus on physical activity. This might involve a more detailed exploration of how restrictions specifically affected older adults’ ability to engage in physical exercise and the subsequent effects on their physical and mental health.

  1. 3- Method (well desighned and written)

  2. -This part needs  detailed breakdown of participants, including age, nationality, and type of physical activity.

  3. 4- Results and discussion: 
  4. It is recommended to consider summarizing the comparison of the study's results with those of other research, placing greater emphasis on the challenges encountered during the study.

Author Response

For reviewer 2

Dear Researcher,

It’s a nice research idea with practical application. some revisions are recommended as follows: 

First of all, I want to thank you for the time you dedicated to reviewing the article and for your comments on how to improve it.

I'm writing to inform you that I have edited and improved the article. All changes have been highlighted in yellow in the document so they are easily recognisable.

The main changes have been to improve and update the bibliographic references, the abstract, the introduction, and the conclusion, as well as to make specific changes to better understand some of the arguments in the result section.

Now, regarding your comments.

1- Abstract

-The abstract begins by stating the duration of the research (nine months) but could benefit from a clearer articulation of the research objectives. It mentions using exploratory-descriptive qualitative research but does not specify the research questions or hypotheses guiding the study.

This work developed an ethnography, and as such, it did not start with a hypothesis or a detailed research question. The general framework of the research, as expressed in the abstract, was to understand the meanings attributed to physical exercise and health by older people (in the context of COVID-19 and post-COVID-19).

-The abstract indicates that the study explores the meanings attributed to physical exercise during and after COVID-19, emphasizing safety, belonging, and meaningfulness. However, it could be strengthened by providing specific examples or key findings that illustrate these themes.

I made some modifications to the abstract to gain more clarity. They can be seen in the text highlighted in yellow.

2- Introduction

-this section needs to be improved. Incorporating theoretical frameworks or concepts that highlight the necessity of conducting such a study would strengthen the introduction. This could involve outlining theories of aging, physical activity, or health behavior change that guide the research and how these theories will be applied or tested in the study. the following updated references are recommended:

 Faraziani, F., & Eken , Özgür . (2024). Enhancing Cognitive Abilities and Delaying Cognitive Decline in the Elderly through Exercise-based Health Management Systems. International Journal of Sport Studies for Health, 7(2), 13-22. 

Ahmadpour, M., & Rezaei, . M. . (2023). The Effect of Whole Body Electromyostimulation Exercises on Improving Static Balance and Self-Efficacy in the Elderly. Health Nexus, 1(4), 39-47. 

The introduction was modified, and the recommended texts were integrated into the article (They can be seen in the text highlighted in yellow).

  1. The introduction thoroughly focus on the impact of COVID-19 on older adults but could more closely tie these impacts to the study’s focus on physical activity. This might involve a more detailed exploration of how restrictions specifically affected older adults’ ability to engage in physical exercise and the subsequent effects on their physical and mental health.

I have done it, which can be seen in the new version of the article.

Method (well desighned and written)

Thank you for your comment.

The methodology needs  detailed breakdown of participants, including age, nationality, and type of physical activity.

The nationality, the age, and the activity carried out are detailed in the table.

Results and discussion:

It is recommended to consider summarizing the comparison of the study's results with those of other research, placing greater emphasis on the challenges encountered during the study.

I have made these suggestions, which can be seen in the new version of the article.

Once again, your feedback has been instrumental in shaping my work, and I'm grateful for the opportunity to address your concerns. I have taken your suggestions and implemented several changes to strengthen my article. Thank you.

Round 2

Reviewer 1 Report

Comments and Suggestions for Authors

Dear author, I appreciate your commitment to consider my comments. However, except for comment 4, in my opinion, the responses do not resolve the suggestions.  I would like to point out that the two suggested references were conceived and indicated in the context of strategies implemented to keep the elderly and subjects with diseases active during COVID-19 and were original articles with precise information on exercise protocols to be possibly repeated. However, the article with the review nature that you included might come closer to that purpose. My doubt remains about the practical implications and innovativeness of the work.

Author Response

Dear reviewer, I have made three changes:

1.- I have integrated the recommended articles.

2.- I have reworked some paragraphs of the conclusion to make it clearer what the contributions of the study are.

3.- I added an idea to the methodology to make the participant recruitment section more robust.

Changes from the first revision are still in yellow, and new changes are in red.

I thank you for taking the time to review my work.

Reviewer 2 Report

Comments and Suggestions for Authors

The revisions are accepted. 

Author Response

Dear reviewer, I'm grateful for your expertise and the time you dedicated to reviewing my article.